# Total Diet Study to Assess Radioactive Cs and ^40^K Levels in the Japanese Population before and after the Fukushima Daiichi Nuclear Power Plant Accident

**DOI:** 10.3390/ijerph17218131

**Published:** 2020-11-03

**Authors:** Hiroshi Terada, Ikuyo Iijima, Sadaaki Miyake, Kimio Isomura, Hideo Sugiyama

**Affiliations:** 1Department of Environmental Health, National Institute of Public Health, Saitama 351-0197, Japan; sugiyama.h.aa@niph.go.jp; 2Chemistry Division, Kanagawa Prefectural Institute of Public Health (Retired), Kanagawa 253-0087, Japan; ikijima@icloud.com; 3Biological Effect Research Group, Saitama Prefectural Institute of Public Health, Saitama 355-0133, Japan; miyake.sadaaki@pref.saitama.lg.jp; 4Hyogo Prefectural Institute of Public Health and Environmental Sciences (Retired), Kobe 652-0032, Japan; zaihuriboku@gmail.com

**Keywords:** total diet study, radioactive cesium, potassium-40, dietary intake, dose assessment, Fukushima accident

## Abstract

We conducted a total diet study (TDS) of ^137^Cs, ^134^Cs, and ^40^K to assess their average dietary exposure levels in a Japanese adult population before and after the Fukushima Daiichi nuclear power plant (FDNPP) accident. Nineteen market baskets were evaluated in 2006–2011. In each basket, a TDS sample comprising tap water and 160–170 food items, which were combined into 13 groups, were collected for analysis by gamma-ray spectrometry. From 2006 to 2010, the ^137^Cs activity concentration in the “fish and shellfish” group was 0.099 Bq/kg, representing the highest value obtained, whereas the total committed effective dose (CED) of radiocesium isotopes (^137^Cs + ^134^Cs) was 0.69 μSv. In 2011, “milk and dairy products” from Sendai City had a Cs activity concentration of 12 Bq/kg, representing the highest values among all food groups studied. However, the annual CED of radioactive Cs in Fukushima City was 17 μSv after the FDNPP accident, which is 60-fold lower than the maximum permissible dose of 1 mSv/year. The mean CED obtained for ^40^K was 180 μSv, which is comparable to the global average. Our results reveal the average dietary exposure of ^137^Cs, ^134^Cs, and ^40^K, which can aid in estimating the radiological safety of foods.

## 1. Introduction

To ensure food safety, it is essential to assess the exposure levels to toxic substances in food. Currently, there are three approaches for estimating the dietary intake of such substances: total diet study (TDS), duplicate portion study (DPS), and selective study of individual foodstuffs. A TDS, also referred to as a market basket study, has an advantage over the two alternatives in terms of its accuracy. Furthermore, TDS takes into account the effect of kitchen preparation on the levels of toxic substances in food, and provides information on which food groups are the main sources of contamination [1]. Therefore, TDSs have been supported and endorsed by the Food and Agriculture Organization of the United Nations (FAO) and the World Health Organization (WHO) since the 1970s. According to a survey carried out in 2015 by Health Canada, in cooperation with the WHO, approximately 53 countries around the world perform TDS analyses [2].

The first TDS was conducted in response to public concerns about a radioactive fallout from atmospheric nuclear testing [3,4]. Fallouts contain hundreds of different radionuclides. Among these, ^137^Cs and ^90^Sr are the most significant sources of internal exposure to radiation, owing to their long half-lives of 30.17 years and 28.8 years, respectively, as well as to their chemical similarities to essential elements (^137^Cs resembles potassium and ^90^Sr resembles calcium). Therefore, the U.S. Food and Drug Administration (FDA) has been carrying out TDSs focusing on these two radionuclides since 1961 [5].

In Japan, TDSs have been performed annually by the Ministry of Health, Labour and Welfare (MHLW) since 1977 [6,7]. Initially, pesticides and their metabolites, seven metals (Pb, Cd, total Hg, total As, Cu, Mn, and Zn), and total polychlorinated biphenyl (PCB) were analyzed in these studies [6,8]. Similarly, radioactivity monitoring has been conducted by the Nuclear Regulation Authority (NRA) since the 1950s. Radionuclide levels in airborne dust, rainwater, river water, seawater, soil, and food have been analyzed in this monitoring [9]. Until 2008, the NRA also conducted DPSs to evaluate the daily dietary intake of radionuclides [10,11,12]. Subsequently, TDSs of radionuclides began in 2003 by the MHLW to assess total exposure levels of radionuclides in the average Japanese diet, as well as the contribution of each food group to this total. Between 2003 and 2011, we carried out a TDS, which was supported by a Health and Labour Sciences Research Grant (HLSRG) from the MHLW [13,14,15,16,17,18,19]. From 2003 to 2005, the dietary exposure to man-made radionuclides (^137^Cs, ^134^Cs, and ^90^Sr) and natural radionuclides (^40^K, ^214^Pb, ^214^Bi, ^228^Ac, ^212^Pb, ^208^Tl, and U) were determined. Sugiyama et al. [13] revealed that only trace amounts of ^137^Cs were found in TDS samples, and that “fish and shellfish” contained 0.145 Bq/kg, representing the highest activity concentration measured. They also found that the daily dietary intake and the committed effective dose (CED) of ^137^Cs were, at the very most, 0.080 Bq/person/day and 0.38 μSv, respectively, with the main sources of the exposure being from the “fish and shellfish”, “meat and eggs”, and “other vegetables, mushrooms and seaweeds” food groups.

The present study aimed to assess trends in dietary exposure levels to γ-emitting radionuclides, namely ^137^Cs, ^134^Cs, and ^40^K, from typical Japanese diets via TDS. The concentrations, dietary intake, and CEDs of these radionuclides from 2006 to 2011 have been presented. The results from before and after the Fukushima Daiichi nuclear power plant (FDNPP) accident that occurred in March 2011 have been compared. We also attempted to summarize the results of the TDS performed between 2012 and 2019 by other institutions in Japan, because almost 10 years have passed since the FDNPP accident.

## 2. Materials and Methods

### 2.1. Sampling and Preparation

Each year, TDS samples have been collected from three or four cities in different regional blocks of Japan, except for the year 2010. Each sample collection is referred to as a market basket (MB). We collected as many locally produced food items as possible. The most preferred production areas in descending order were same prefecture, same regional block, and other regional blocks in Japan. As shown in Figure 1, 19 MBs were collected from all regional blocks of Japan, other than Minami-Kyushu, from 2006 to 2011. In each MB, 160 to 170 food items were purchased from local retailers. These items were classified into 13 food groups as follows: rice and rice products (Group I); cereals, seeds, nuts, and potatoes (Group II); sugars and confectionaries (Group III); fats and oils (Group IV); pulses and their products (Group V); fruits (Group VI); green and yellow vegetables (Group VII); other vegetables, mushrooms, and seaweeds (Group VIII); beverages (Group IX); fish and shellfish (Group X); meat and eggs (Group XI); milk and dairy products (Group XII); and seasonings (Group XIII). The relative proportion of each food item within a group was based on average regional food consumption data for individuals 20 years and over, which were obtained from the National Health and Nutrition Survey (NHNS) performed by the MHLW between 2002 and 2004. Individual food items were cooked, if necessary, and then combined for analysis into 13 food groups. In addition, tap water was collected in each MB for drinking water (Group XIV). Thus, in total 19 × 14 = 266 TDS samples were obtained. The sample weights of individual food groups were approximately 5 kg, with the exception of 12 kg for “rice and rice products” and 100 kg for “drinking water”.

TDS samples, other than “fats and oils” and “drinking water”, were freeze-dried or heat-dried and then incinerated at 450 °C for 24 h into ash. The ash samples were placed in separate plastic containers with a capacity of 100 mL. The “drinking water” samples were condensed by heating and evaporated to dryness, and the residues were stored in the plastic containers, as described above. The “fats and oil” samples were stored in 1 L Marinelli beakers in their raw state.

### 2.2. Determination of ^137^Cs, ^134^Cs, and ^40^K

Cs-137, Cs-134, and K-40 in the TDS samples were detected for a minimum of 80,000 s using high-purity Ge semiconductor detectors (2519 and GC2018 of CANBERRA Co., Meriden, CT, USA; EGPC 20-190-R of EURYSIS Co., Lingolsheim, Cedex-France; CNVD30-35195 of OXFORD Co., Oxon, Oxford, UK; and IGC40200 and IGC25190SD of PGT Co., Princeton, NJ, USA). Activity concentrations of the three radionuclides were corrected as of the end of the sampling period and expressed in Bq/kg of fresh weight. Energy and efficiency calibrations were performed using γ-ray volume sources (MX033U8PP and MX033MR, Japan Radioisotope Association, Tokyo, Japan). The limits of detection (LODs) for ^137^Cs and ^134^Cs were approximately 0.05 Bq/kg for all food groups, except for “fats and oils” and “seasonings”.

### 2.3. Calculation of Dietary Intake and the Committed Effective Dose (CED) Values

The daily dietary intake of radioactive Cs and ^40^K for adults was calculated from the activity concentrations of the three radionuclides in the samples, as described above. The radionuclide activity concentration was multiplied by the average food consumption value for the adult population, as determined by the NHNS, in each regional block to provide the radionuclide intake for each radionuclide and food group. Next, the calculated intake for each food group was summed to estimate the total dietary intake of a given radionuclide in each MB. The dietary intake of radioactive Cs was obtained by summation of ^137^Cs and ^134^Cs intake values. If a target radionuclide was not detected in a sample, the activity concentration of the non-detected (ND) radionuclide was assumed to be zero or the LOD value. The lower and upper limits of the total radionuclide intake values were estimated by assuming the ND value as zero and the LOD, respectively.

The CED values associated with Cs and ^40^K for adults were estimated under the assumption of a one-year intake of the TDS samples, and were obtained by applying the dietary radionuclide intake values and dose coefficients for ingestion of each radionuclide, given by the International Commission on Radiological Protection (ICRP), to be 1.3 × 10^−8^ Sv/Bq for ^137^Cs, 1.9 × 10^−8^ Sv/Bq for ^134^Cs, and 4.6 × 10^−9^ Sv/Bq for ^40^K [20]. The CED of radioactive Cs was calculated by summing those of ^137^Cs and ^134^Cs. The ND radionuclides were given a value of zero and the LOD for estimating the lower and upper limits of the CED values, respectively.

## 3. Results and Discussion

### 3.1. Levels of ^137^Cs, ^134^Cs, and ^40^K

#### 3.1.1. ^137^Cs Levels before the FDNPP Accident

We collected 16 MBs from 13 cities between 2006 and 2010, resulting in a total of 224 TDS samples. Cesium-137 was the only anthropogenic radionuclide observed in these samples, as ^134^Cs, which was released from the Chernobyl disaster, was not detected at all. Table 1 summarizes the ^137^Cs activity concentrations in TDS samples from 2006 to 2010. The activity concentrations of ^137^Cs in TDS samples were all below 0.1 Bq/kg during this period. The food group “fish and shellfish” had a ^137^Cs activity concentration of 0.099 Bq/kg, representing the highest value measured, followed by “other vegetables, mushrooms and seaweeds” (0.092 Bq/kg), “meat and eggs” (0.083 Bq/kg), and “milk and dairy products” (0.057 Bq/kg). The highest mean ^137^Cs activity concentrations were found in “fish and shellfish” (0.072 Bq/kg) followed by “other vegetables, mushrooms and seaweeds” (0.024 Bq/kg), “meat and eggs” (0.024 Bq/kg), and “milk and dairy products” (0.022 Bq/kg). These results are similar to the TDS carried out between 2003 and 2005 [13]. Cesium-137 was detected in all “fish and shellfish” samples, whereas it was not found in any samples of “fats and oils”, “seasonings”, and “drinking water”. The detection rate of ^137^Cs was about 30%, and was the highest in “fish and shellfish” (100%), followed by “milk and dairy products” (68.8%), “meat and eggs” (62.5%), and “other vegetables, mushrooms and seaweeds” (50.0%).

#### 3.1.2. ^137^Cs and ^134^Cs Levels after the FDNPP Accident

We obtained 42 TDS samples from Sendai City, Fukushima City, and Tokyo in October and November of 2011, approximately six months after the FDNPP accident. According to the United Nations Scientific Committee on the Effects of Atomic Radiation (UNSCEAR) [21], the total release of ^137^Cs, ^134^Cs, and ^131^I in the atmosphere resulting from the FDNPP incident was estimated to be 8.8 PBq, 9.0 PBq, and 124 PBq, respectively. Immediately after the FDNPP accident, some foodstuffs, such as spinach and milk, were highly contaminated with ^131^I, which accumulates in the thyroid gland and may increase the risk of thyroid cancer [22,23,24]. However, ^131^I levels decreased rapidly and became undetectable [25,26,27], owing to its short half-life of 8.02 days. Therefore, ^131^I was not observed in the present study.

Cesium-134 was not detected in any of the TDS samples between 2006 and 2010, but was observed in all food groups, with the exception of “fats and oils”, in 2011 as a consequence of the FDNPP accident. Table 2 shows the ^137^Cs and ^134^Cs activity concentration of the TDS samples after the FDNPP accident. The detection rates of ^137^Cs and ^134^Cs were 88% and 69%, respectively. No sample exceeded the ^137^Cs level of 0.1 Bq/kg during 2006 to 2010, whereas 19 samples exceeded this level in 2011. The activity concentrations of ^137^Cs (Bq/kg) were in the range of <0.041–6.400 in Sendai City, <0.089–4.100 in Fukushima City, and <0.045–2.100 in Tokyo; while those of ^134^Cs (Bq/kg) were <0.011–5.400 in Sendai City, <0.040–3.500 in Fukushima City, and <0.025–1.700 in Tokyo. The activity concentration ratios of ^134^Cs to ^137^Cs in the TDS samples were ranged from 0.38 to 0.94, and the mean and standard deviation values were calculated to be 0.73 and 0.14, respectively. 

The highest activity concentration of radioactive Cs (^137^Cs + ^134^Cs) among the 42 TDS samples was 12 Bq/kg, and found in the “milk and dairy products” group from Sendai City. This food group also showed the highest Cs activity concentration in Tokyo (3.8 Bq/kg), while fruits carried the highest activity concentration in Fukushima City (7.6 Bq/kg). Cs activity concentrations in the fruits of Fukushima City were characteristically higher than those in Sendai City (0.093 Bq/kg) and Tokyo (0.17 Bq/kg). Although “green and yellow vegetables”, such as spinach, showed extremely high Cs activity concentrations, with values exceeding 10,000 Bq/kg immediately following the FDNPP accident, their levels declined rapidly [26,27,28]. Thus, Cs activity concentrations in “green and yellow vegetables” were at most 0.58 Bq/kg in our TDS samples. The “meat and eggs” group was found to have higher Cs activity concentration before the FDNPP accident, while the relatively low activity concentration after the accident is due to the import of animal feed from foreign countries. Interestingly, ^137^Cs and ^134^Cs were not found in any “fats and oil” samples before or after the accident. This may be due to their poor solubility in lipids.

After the FDNPP accident, regulation of the levels of radionuclides in food was established by the MHLW. On 17 March 2011, the provisional regulation value (RPV) for radioactive Cs was established to be 200 Bq/kg for “drinking water” and “milk and dairy products”, and 500 Bq/kg for vegetables, grain, meat, eggs, and fish [28,29]. Presently, the standard limits, which were put into place on 1 April 2012, are 100 Bq/kg for general foods, 50 Bq/kg for milk and infant foods, and 10 Bq/kg for drinking water [30,31]. In the present study, although Cs activity concentrations of the TDS samples increased significantly after the FDNPP accident, the concentrations were well below the regulatory levels. The highest radioactive Cs value of 12 Bq/kg in this TDS was 17 times lower than the RPV for “milk and dairy products”, and four times lower than the standard limit for milk.

#### 3.1.3. ^40^K Levels

Potassium is an essential element in food products and humans. Potassium-40, whose natural abundance is 0.0117%, is the only naturally occurring radionuclide of potassium, and has a very long half-life of 1.248 billion years. Accordingly, ^40^K in foods contributes considerably to internal exposure in the general public. The activity concentration values of ^40^K in the TDS samples are listed in Table 3. In contrast to ^137^Cs and ^134^Cs, ^40^K was detected in all food groups, and the levels were in the range of 10–100 Bq/kg in most of the TDS samples analyzed. The mean ^40^K level values between 2006 and 2011 measured for individual food groups were the highest in “green and yellow vegetables”, at 90 Bq/kg, followed by “pulses and their products” (84 Bq/kg), and “fish and shellfish” (83 Bq/kg). In contrast, mean ^40^K values were relatively low in “fats and oils”, “drinking water”, “rice and rice products”, and “beverages”. The ^40^K activity concentration was not significantly different before and after the FDNPP accident.

### 3.2. Dietary Intake and CED Values of Radioactive Cs

#### 3.2.1. Before the FDNPP Accident

Table 4 presents the total daily dietary intake values and CEDs of radioactive Cs and ^40^K for adults in each MB. Between 2006 and 2010, the lower and upper limits of radioactive Cs intake varied from 0.0047 to 0.0320 Bq/person/day and from 0.060 to 0.122 Bq/person/day, respectively. The mean values obtained for the lower and upper limits during the same period were 0.020 Bq/person/day (SD = 0.0082) and 0.085 Bq/person/day (SD = 0.019), respectively. Similar results were obtained in a previous study [13]. The lower and upper limits in each MB were significantly different because of the low detection rates of ^137^Cs and ^134^Cs before the FDNPP accident (30% and 0%, respectively).

As previously described, the NRA conducted DPS studies of radionuclides until 2008, and compiled the results from 47 prefectures in Japan into a database [32]. According to the database, the daily intake of ^137^Cs for adults between 2003 and 2008 ranged from below the detection limit to 0.56 Bq/person/day, and the mean was calculated to be 0.018 Bq/person/day (SD = 0.031), assuming that the activity concentration value of ND radionuclides was zero. Although a larger variation was observed, this finding is similar to the mean lower limit of radioactive Cs intake obtained in the present study.

The lower and upper limits of the CEDs of radioactive Cs were in the range of 0.022–0.150 μSv and 0.33–0.69 μSv, with mean values of 0.098 μSv (SD = 0.041) and 0.47 μSv (SD = 0.11), respectively. The current standard limits for radionuclides in food in Japan were established on the basis of an annual CED of 1 mSv, representing the maximum permissible dose due to food consumption [27,31]. Although the upper limit of the CEDs obtained were overestimated, due to the low detection rate of ^137^Cs and ^134^Cs, the highest CED of 0.69 μSv is approximately 1400 times lower than the maximum permissible dose, and thus would seem to pose no health risk.

#### 3.2.2. After the FDNPP Accident

Since ^137^Cs and ^134^Cs were detected in most of the food groups after the FDNPP accident, the lower and upper limits of radioactive Cs intakes were calculated using actual activity concentrations of these radionuclides, rather than zero or LODs. Thus, the lower and upper limits were almost the same in each MB after the FDNPP accident, while they were significantly different before the accident. The lower limits of total radioactive Cs intake were estimated to be 2.9 Bq/person/day in Fukushima City, 2.2 Bq/person/day in Sendai City, and 0.67 Bq/person/day in Tokyo, after the FDNPP accident (Table 4).

The lower limits of the total CED of radioactive Cs in 2011 were 12 μSv in Sendai City, 17 μSv in Fukushima City, and 3.8 μSv in Tokyo. The high value obtained for Fukushima City is expected, as it is located the closest to the FDNPP incident. Figure 2 displays the lower limits of CED values of radioactive Cs from 2003 to 2011, and shows that values obtained after the FDNPP accident were about two orders of magnitude higher than before the accident. Nevertheless, the highest CED of radioactive Cs (17 μSv), which was the exposure level in half a year after the FDNPP accident, is approximately 60 times lower than the maximum permissible dose.

Figure 3 summarizes the CED values of radioactive Cs, organized by food group, in 2011. The three food groups with the highest contributions to the CEDs were “milk and dairy products” (66%), “fish and shellfish” (17%), and “rice and rice products” (8.2%) in Sendai City; “rice and rice products” (33%), fruits (32%), and “milk and dairy products” (15%) in Fukushima City; and “milk and dairy products” (69%), “rice and rice products” (7.6%), and “fish and shellfish” (7.0%) in Tokyo.

In 2011, TDSs of radioactive Cs and ^40^K were also carried out by Tsutsumi et al. [33] and Miyazaki et al. [34]. Tsutsumi et al. analyzed MBs from Sendai City, Fukushima City, and Tokyo in September and November 2011, and revealed that the lower limits of total CED values of radioactive Cs were 17 μSv in Sendai City, 19 μSv in Fukushima City, and 2.1 μSv in Tokyo. They also found that “fish and shellfish”, “fruits”, “green and yellow vegetables”, and “milk and dairy products” were the main contributors to the CED values obtained. Although the contributing food groups varied to some extent, their findings are similar to the total CED values of radioactive Cs obtained in the present study. Miyazaki et al. collected an MB from Nagoya City in August 2011, and measured CED values for radioactive Cs of 1.5 μSv, which was lower than that obtained in both our study and that of Tsutsumi et al. This might be because of the distance between the FDNPP accident site and Nagoya City, which at 446 km, is much larger than that to Sendai City (95 km), Fukushima City (62 km), or Tokyo (227 km).

After 2011, the TDSs have been performed by other institutions in Japan. The National Institute of Health Sciences have been conducting the TDS, which was targeting 15 areas, including Fukushima Prefecture, Miyagi Prefecture, and Tokyo, twice a year since 2011. Uekusa et al. [35] revealed that the maximum CEDs due to radioactive Cs decreased to 9.4 μSv (March 2012), 3.8 μSv (September 2012), and 7.1 μSv (March 2013) from 19 μSv in 2011 [33]. In the past two years, the results were as follows: 1.1 μSv (February–March 2018) [36], 1.1 μSv (September–October 2018) [37], 1.0 μSv (February–March 2019) [38], and 1.0 μSv (September–October 2019) [39]. The Tokyo Metropolitan Institute of Public Health also have been carrying out the TDS in Tokyo once a year. Their results between 2012 and 2019 showed that the CEDs due to radioactive Cs in Tokyo ranged from 0.20 μSv (2017) to 1.3 μSv (2012) [40]. All the results mentioned above indicated that the CEDs due to radioactive Cs clearly decreased from those in 2011, as expected. In 2019, the maximum CED of 1.0 μSv was 17 times lower than that from the present study, and was one-thousandth of the maximum permissible level. However, the maximum CED between 2012 and 2019 was still higher than the maximum upper limit of the CED, due to radioactive Cs before the FDNPP accident (0.69 μSv).

### 3.3. Dietary Intake and CED Values of ^40^K

The minimum and maximum total dietary intake values of ^40^K between 2006 and 2011 were 65 Bq/person/day and 94 Bq/person/day, respectively (Table 5), and the mean was calculated to be 79 Bq/person/day (SD = 7.0). Since ^40^K was found in almost all TDS samples, the upper limit was in accordance with the lower limit in each MB. The intake values of ^40^K showed smaller variation than those obtained for Cs, and no significant difference was observed before or after the FDNPP accident.

The present study was based on food consumption data derived from the NHNS between 2002 and 2004, as previously described. According to the NHNS, the daily K intake for adults was 2.452 g/person/day in 2002 [41], 2.426 g/person/day in 2003 [42], and 2.372 g/person/day in 2004 [43]. Since 1 g of K corresponds to 30.3 Bq of ^40^K, the intake of this radionuclide can be estimated to be 74 Bq/person/day in 2002, 74 Bq/person/day in 2003, and 72 Bq/person/day in 2004; this is in close agreement with the mean ^40^K intake value obtained in the present study. On the other hand, ^40^K intake between 2003 and 2008 varied from 5 to 150 Bq/person/day, according to a DPS conducted by the NRA [32]. The mean ^40^K intake was calculated to be 57 Bq/person/day (SD = 16), and was clearly different from our finding. The DPS was performed on five consecutive days from June to January, whereas the NHNS was conducted for only a single day in November, to ensure the participation of a sufficient number of households. Thus, it should be noted that seasonal changes in food item availability and individual daily variations in food intake cannot be assessed using the NHNS and this TDS [7,44].

The minimum, maximum, and mean total CED values of natural occurring radionuclide ^40^K between 2006 and 2011 were 150 μSv, 210 μSv, and 180 μSv (SD = 16), respectively, which are in close agreement with a previous TDS [13]. The mean value obtained was approximately ten times higher than the maximum CED obtained for artificial radioactive Cs in the present study, and was comparable to the worldwide average of 170 μSv [45]. Figure 4 shows the percent contributions of the contributing food groups to the CED values of ^40^K in 2011. The food group “other vegetables, mushrooms, and seaweeds” had the greatest contribution to the CED values in Sendai City (31%), Fukushima City (26%), and Tokyo (32%). These results were similar to those of Tsutsumi et al. [33] and Miyazaki et al. [34].

### 3.4. TDS of Radionuclides Conducted by Other Countries

The second International Workshop on Total Diet Studies compiled a list of core (screening), intermediate, and comprehensive (refined assessment) priority chemicals that should be considered for inclusion in a TDS [46]. Radionuclides are categorized into the intermediate list, and there are three countries that routinely conduct TDSs, including Japan [2].

The United States has been performing TDSs continuously since 1961, as mentioned previously. The FDA routinely analyzes the following radionuclides in their TDS analysis: ^137^Cs, ^90^Sr, ^106^Ru, ^131^I, and ^40^K. According to the FDA’s report [47], the following three of 2984 samples were above the reporting limit of 5 Bq/kg for ^137^Cs between 2006 and 2014: “baby foods, squash” in 2007 (93.3 Bq/kg), “raisin bran cereal” in 2009 (10.8 Bq/kg), and “salad dressing, creamy/buttermilk type, low calorie” in 2014 (40.5 Bq/kg).

Canada also continuously carries out TDSs of radionuclides. Health Canada has been monitoring natural (^40^K and ^210^Pb) and artificial (^137^Cs, ^134^Cs, ^131^I, ^241^Am, ^57^Co, and ^60^Co) radionuclides in foods since 2000. Their results between 2015 and 2017 showed that all 480 samples tested contained ^137^Cs and ^134^Cs levels below the minimum detection limit, showing a value of approximately 1.3 Bq/kg [48].

The result obtained from the American study was comparable to our finding, in that only one of the 266 TDS samples exceeded ^137^Cs levels of 5 Bq/kg. Similarly, the Canadian study was consistent with respect to ^137^Cs levels, specifically for those observed before the FDNPP accident in the present study. However, such a comparison should be interpreted with caution, since the number of composite samples analyzed in each MB differed between the three countries (approximately 280 for the United States, approximately 160 for Canada, and 14 for the present study) [5,49]. In the present study, individual food items with high levels of ^137^Cs may be underrepresented, due to the grouping of a greater number of food items within each category, which could have lower levels of the radionuclide [1,50]. This dilution effect might have caused the ^137^Cs levels after the FDNPP accident in the present study to be lower than those of “baby foods, squash”, etc., as mentioned above. Meanwhile, the detection rate of ^137^Cs in the present study was much higher than that in the United States and Canada, owing to the lower LOD values of ^137^Cs. The present study evaluated the exposure level to ^137^Cs as previously described, whereas the United States and Canada did not, because ^137^Cs was not detected in nearly any samples.

China [51] and Lebanon [52] also conducted TDSs of radionuclides in 1990 and 2004, respectively. For each MB, both countries collected 12 composite samples, similar to the approach used in the present study. Cs-137 activity concentration in their TDS was found to be below 0.1 Bq/kg for most of the food groups, with the exception of potatoes, which had a value of 10.21 Bq/kg in China; that is consistent with our findings prior to the FDNPP accident.

## 4. Conclusions

The present study provides an estimate of the average dietary exposure of ^137^Cs, ^134^Cs, and ^40^K for adults in Japan. Before the FDNPP accident, activity concentrations of ^137^Cs were in the range of those reported in other countries, and dietary intake values and CEDs were consistent with our previous TDS. Similarly, after the FDNPP accident, the activity concentration and exposure levels of radioactive Cs were well below the regulatory levels, despite an increase in two orders of magnitude being measured. The exposure levels of ^40^K did not differ before and after the FDNPP accident. The mean CED of ^40^K was comparable to the international average, and was 10 times higher than the highest CED of radioactive Cs obtained in the present study.

On an international level, TDSs of radionuclides are scarce, and currently, only Japan evaluates trends in exposure levels to these contaminants. Thus, our findings provide invaluable information with regard to the radiological safety of foods. However, the present study has some limitations. First, the exposure levels immediately after the FDNPP accident could not be estimated, because the TDS in 2011 was performed approximately six months after the accident. Second, seasonal variation cannot be assessed, as the NHNS and this TDS were performed in only one season each year. Finally, the present study did not examine the exposure of children, or exposure to other radionuclides, such as ^90^Sr, Pu, and ^106^Ru, which are targets of standard limit regulations in foods. Therefore, further studies are required to determine the dietary exposure levels of these radionuclides for both adults and children.

## Figures and Tables

**Figure 1 ijerph-17-08131-f001:**
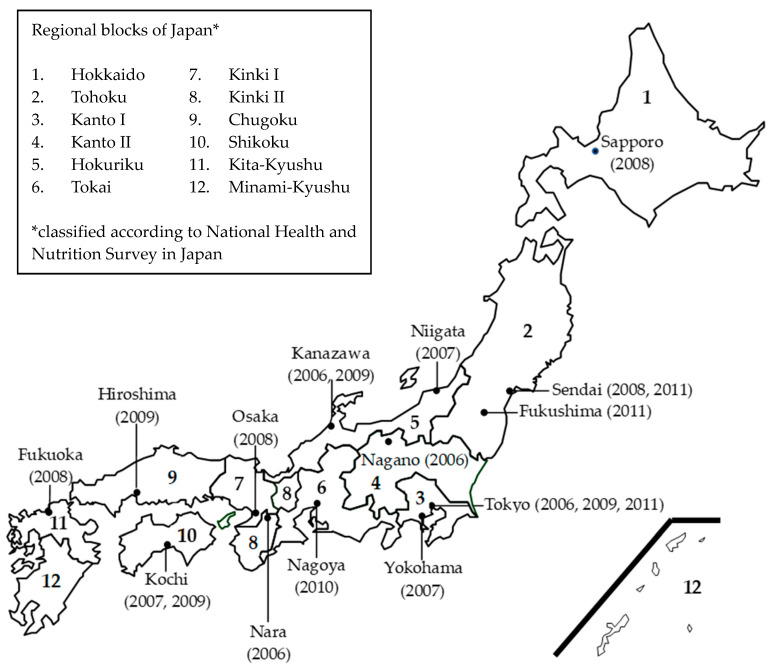
Sampling sites of included in the total diet study conducted from 2006 to 2011. Sampling year of each site is denoted in parenthesis.

**Figure 2 ijerph-17-08131-f002:**
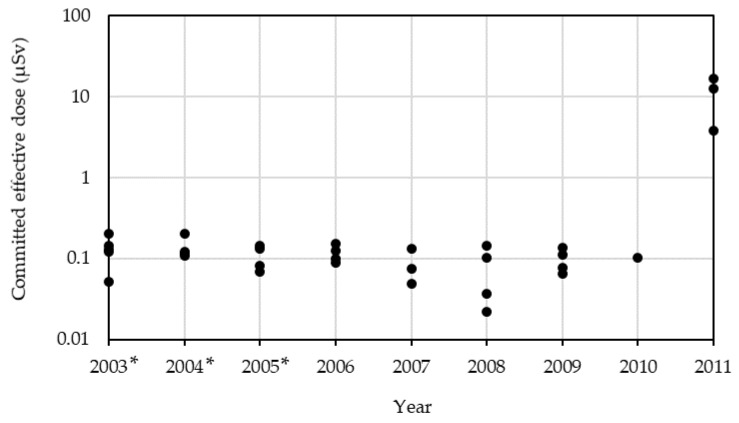
Committed effective doses of radioactive Cs (^137^Cs + ^134^Cs) obtained from each market basket (2003–2011). The closed circles show the results of each market basket. The doses were estimated with the assumptions of one-year intake values of the total diet samples, and the activity concentration values of non-detected radionuclides to be zero.

**Figure 3 ijerph-17-08131-f003:**
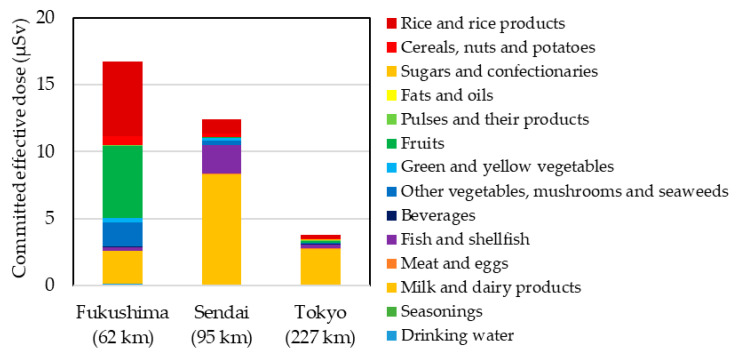
Committed effective doses of radioactive cesium (^137^Cs + ^134^Cs) by food groups in 2011. Distance from the Fukushima Daiichi nuclear power plant to each city is denoted in parentheses. This figure was modified from [19].

**Figure 4 ijerph-17-08131-f004:**
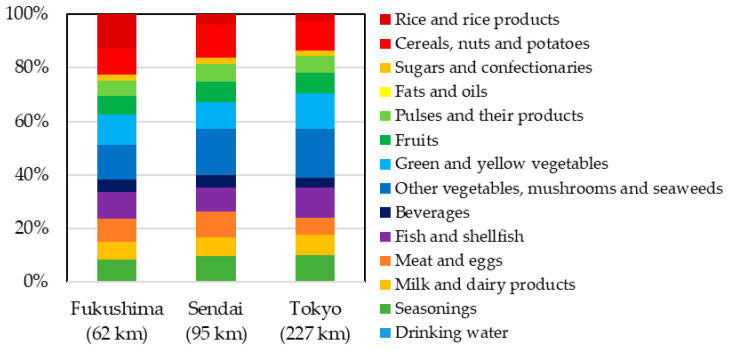
Percentage contributions of food groups to the committed effective doses of ^40^K in 2011. Distance from the Fukushima Daiichi nuclear power plant to each city is denoted in parentheses. This figure was modified from [19].

**Table 1 ijerph-17-08131-t001:** Activity concentrations of ^137^Cs in Japanese total diet study samples from 2006 to 2010.

Food Group		Activity Concentrations of ^137^Cs (Bq/kg)									Mean ± SD *^3^	Percentage of Detection (%) *^4^
		2006 (*n* = 4)		2007 (*n* = 3)		2008 (*n* = 4)		2009 (*n* = 4)		2010 (*n* = 1)		
		Min. *^1^	Max. *^2^	Min. *^1^	Max. *^2^	Min. *^1^	Max. *^2^	Min. *^1^	Max. *^2^			
I	Rice and rice products	<0.003	0.004	<0.005	0.0045	<0.004	0.0061	<0.008	0.013	0.020	0.0036 ± 0.0059	37.5
II	Cereals, nuts, and potatoes	<0.010	<0.025	<0.015	0.016	<0.014	0.034	<0.009	0.033	<0.010	0.0058 ± 0.012	25.0
III	Sugars and confectionaries	<0.018	0.035	<0.012	<0.052	<0.015	0.019	<0.022	<0.038	<0.150	0.0049 ± 0.011	18.8
IV	Fats and oils	<0.032	<0.056	<0.024	<0.049	<0.014	<0.065	<0.022	<0.078	<0.220	0	0
V	Pulses and their products	<0.040	<0.058	<0.024	0.016	<0.024	0.019	<0.014	0.032	<0.076	0.0042 ± 0.010	18.8
VI	Fruits	<0.010	<0.024	<0.010	0.027	<0.011	<0.037	<0.010	0.038	<0.026	0.0041 ± 0.011	12.5
VII	Green and yellow vegetables	<0.022	0.015	<0.016	0.018	<0.009	<0.039	<0.024	<0.029	<0.024	0.0021 ± 0.0057	12.5
VIII	Other vegetables, mushrooms, and seaweeds	<0.016	0.092	<0.029	0.069	<0.015	0.031	<0.022	0.024	<0.018	0.024 ± 0.031	50
IX *^5^	Beverages	<0.018	<0.024	<0.004	<0.015	<0.007	0.0073	<0.004	0.0049	<0.006	0.00077 ± 0.00210	12.5
X	Fish and shellfish	0.067	0.099	0.078	0.093	0.029	0.071	0.062	0.093	0.048	0.0720 ± 0.0205	100.0
XI	Meat and eggs	<0.023	0.041	<0.018	0.032	<0.020	0.083	<0.025	0.054	0.028	0.024 ± 0.024	62.5
XII	Milk and dairy products	<0.021	0.057	<0.025	0.018	<0.027	0.033	<0.042	0.051	0.035	0.022 ± 0.012	68.8
XIII *^6^	Seasonings	<0.012	<0.140	<0.075	<0.155	<0.054	<0.14	<0.028	<0.087	<0.089	0	0
XIV	Drinking water	<0.0002	<0.0004	<0.0001	<0.0004	<0.0001	<0.0005	<0.0001	<0.0003	<0.0002	0	0

*n* = number of the market basket(s) performed in each year. *^1^ Minimum ^137^Cs activity concentration in each year. *^2^ Maximum ^137^Cs activity concentration in each year. *^3^ Mean and standard deviation values of ^137^Cs activity concentrations, which were calculated under the assumption that the concentrations of the non-detected were zero. *^4^ Detection rate of ^137^Cs for each food group. *^5^ This food group was categorized as “Seasonings and beverages” in 2006. *^6^ This food group was categorized as “Others” in 2006.

**Table 2 ijerph-17-08131-t002:** Activity concentrations of ^137^Cs and ^134^Cs in Japanese total diet study samples in 2011.

Food Group		Activity Concentrations of ^137^Cs (Bq/kg)			Activity Concentrations of ^134^Cs (Bq/kg)			Activity Concentrations of ^137^Cs + ^134^Cs (Bq/kg)		
		Sendai	Fukushima	Tokyo	Sendai	Fukushima	Tokyo	Sendai	Fukushima	Tokyo
I	Rice and rice products	0.270	1.500	0.079	0.210	1.200	0.057	0.480	2.600	0.140
II	Cereals, nuts, and potatoes	0.170	0.340	0.058	0.140	0.320	0.035	0.310	0.660	0.093
III	Sugars and confectionaries	0.120	0.180	0.053	0.054	0.130	<0.025	0.170	0.320	0.053
IV	Fats and oils	<0.087	<0.089	<0.045	<0.072	<0.100	<0.048	<0.160	<0.190	<0.093
V	Pulses and their products	<0.041	0.041	0.130	<0.025	<0.040	0.098	<0.066	0.041	0.230
VI	Fruits	0.0570	4.100	0.093	<0.036	3.500	0.074	<0.093	7.600	0.170
VII	Green and yellow vegetables	0.200	0.310	0.059	0.160	0.270	0.025	0.360	0.580	0.084
VIII	Other vegetables, mushrooms, and seaweeds	0.150	0.710	0.044	0.084	0.580	<0.028	0.230	1.300	0.044
IX *^1^	Beverages	0.017	0.015	0.018	<0.011	0.011	0.012	<0.028	0.026	0.030
X	Fish and shellfish	2.200	0.300	0.300	1.800	0.210	0.210	4.000	0.510	0.510
XI	Meat and eggs	0.074	0.063	0.056	0.047	<0.040	0.021	0.120	0.063	0.077
XII	Milk and dairy products	6.400	2.000	2.100	5.400	1.600	1.70	12.000	3.500	3.800
XIII *^2^	Seasonings	<0.230	0.052	0.140	<0.160	<0.088	<0.110	<0.390	0.052	0.140
XIV	Drinking water	0.0087	0.017	0.0048	0.0071	0.015	0.0038	0.016	0.032	0.0086

*^1^ This food group was categorized as “Seasonings and beverages” in 2006. *^2^ This food group was categorized as “Others” in 2006.

**Table 3 ijerph-17-08131-t003:** Activity concentrations of ^40^K in Japanese total diet study samples from 2006 to 2011.

Food Group		Activity Concentrations of ^40^K (Bq/kg) *^1^									Mean ± SD *^4^ (Bq/kg)	Activity Concentrations of ^40^K (Bq/kg) in 2011 *^5^		
		2006 (*n* = 4)		2007 (*n* = 3)		2008 (*n* = 4)		2009 (*n* = 4)		2010 (*n* = 1)				
		Min. *^2^	Max. *^3^	Min. *^2^	Max. *^2^	Min. *^2^	Max. *^3^	Min. *^2^	Max. *^3^			Fukushima	Sendai	Tokyo
I	Rice and rice products	5.1	7.1	5.7	9.7	7.1	7.9	5.7	11.0	3.5	7.1 ± 1.7	31.0	8.3	5.4
II	Cereals, nuts, and potatoes	42.0	53.0	44.0	63.0	48.0	55.0	50.0	57.0	48.0	51.0 ± 5.1	52.0	62.0	54.0
III	Sugars and confectionaries	23.0	44.0	54.0	64.0	65.0	71.0	51.0	64.0	67.0	56.0 ± 14	65.0	67.0	49.0
IV	Fats and oils	<1.2	4.3	<0.81	0.9	<0.90	1.6	<0.22	1.2	<2.7	1.0 ± 1.3 *^6^	<1.2	<1.2	<1.1
V	Pulses and their products	94.0	104.0	70.0	103.0	72.0	98.0	68.0	96.0	59.0	85.0 ± 15	78.0	80.0	76.0
VI	Fruits	41.0	54.0	46.0	57.0	48.0	55.0	44.0	61.0	61.0	52.0 ± 5.8	50.0	51.0	48.0
VII	Green and yellow vegetables	76.0	99.0	63.0	106.0	85.0	104.0	78.0	99.0	95.0	89.0 ± 11	95.0	79.0	103.0
VIII	Other vegetables, mushrooms, and seaweeds	47.0	106.0	93.0	136.0	47.0	105.0	39.0	103.0	55.0	81.0 ± 29	48.0	58.0	59.0
IX *^7^	Beverages	23.0	28.0	7.9	9.0	7.4	11.0	3.2	9.3	8.1	12.0 ± 8.2	8.6	7.8	6.3
X	Fish and shellfish	74.0	95.0	81.0	88.0	70.0	87.0	75.0	99.0	77.0	82.0 ± 7.9	96.0	79.0	96.0
XI	Meat and eggs	64.0	89.0	68.0	85.0	61.0	80.0	60.0	82.0	77.0	73.0 ± 9.8	73.0	74.0	47.0
XII	Milk and dairy products	45.0	51.0	41.0	50.0	37.0	43.0	44.0	105.0	43.0	49.0 ± 15	48.0	46.0	51.0
XIII *^8^	Seasonings	11.0	74.0	66.0	88.0	64.0	71.0	28.0	69.0	79.0	64.0 ± 19	76.0	82.0	81.0
XIV	Drinking water	0.019	0.061	0.015	0.05	0.017	0.087	0.021	0.040	0.039	0.041 ± 0.020	0.012	0.018	0.053

*n* = number of the market basket(s) performed in each year. *^1^ Before the Fukushima Daiichi nuclear power plant accident. *^2^ Minimum ^40^K activity concentration in each year. *^3^ Maximum ^40^K activity concentration in each year. *^4^ Mean and standard deviation values of ^40^K activity concentration before the Fukushima Daiichi nuclear power plant accident. *^5^ After the Fukushima Daiichi nuclear power plant accident. *^6^ Calculated under the assumption that the ^40^K activity concentrations of the non-detected were zero. *^7^ This food group was categorized as “Seasonings and beverages” in 2006. *^8^ This food group was categorized as “Others” in 2006.

**Table 4 ijerph-17-08131-t004:** Total daily dietary intake values and committed effective doses of radioactive Cs (^137^Cs + ^134^Cs) for adults in Japan, estimated via total diet studies from 2006 to 2011.

Year		Region	City	Daily Intake (Bq/Person·Day)		Committed Effective Dose (μSv)	
				Lower Limit *^1^	Upper Limit *^2^	Lower Limit *^1^	Upper Limit *^2^
Before the FDNPP *^3^ accident	2006	Kanto I	Tokyo	0.0320	0.079	0.150	0.40
		Kanto II	Nagano	0.0270	0.081	0.150	0.40
		Hokuriku	Kanazawa	0.0190	0.061	0.130	0.45
		Kinki II	Nara	0.0210	0.065	0.089	0.34
	2007	Kanto I	Yokohama	0.0280	0.120	0.099	0.37
		Hokuriku	Niigata	0.0160	0.064	0.130	0.69
		Shikoku	Kochi	0.0100	0.100	0.074	0.37
	2008	Hokkaido	Sapporo	0.0220	0.087	0.049	0.58
		Tohoku	Sendai	0.0047	0.097	0.100	0.58
		Kinki I	Osaka	0.0310	0.060	0.022	0.55
		Kita-Kyushu	Fukuoka	0.0079	0.086	0.140	0.33
	2009	Kanto I	Tokyo	0.0290	0.071	0.037	0.51
		Hokuriku	Kanazawa	0.0230	0.100	0.140	0.40
		Chugoku	Hiroshima	0.0140	0.098	0.110	0.63
		Shikoku	Kochi	0.0160	0.072	0.064	0.55
	2010	Tokai	Nagoya	0.0220	0.120	0.077	0.41
Mean ± SD *^4^				0.0200 ± 0.0083	0.085 ± 0.019	0.098 ± 0.041	0.47 ± 0.11
After the FDNPP *^3^ accident	2011	Tohoku	Sendai	2.20	2.20	12.0	13.0
		Tohoku	Fukushima	2.90	2.90	17.0	17.0
		Kanto I	Tokyo	0.67	0.69	3.8	3.9

*^1^ Assuming that the activity concentrations of the non-detected radionuclides were zero. *^2^ Assuming that the activity concentrations of non-detected radionuclides were equal to the limits of detection. *^3^ FDNPP: Fukushima Daiichi nuclear power plant. *^4^ Mean and standard deviation values before the Fukushima Daiichi nuclear power plant accident.

**Table 5 ijerph-17-08131-t005:** Total daily dietary intake values and committed effective doses of ^40^K for adults in Japan, estimated via total diet studies from 2006 to 2011.

Year	Region		City	Daily Intake (Bq/Person·Day)	Committed Effective Dose (μSv)
Before the FDNPP *^1^ accident	2006	Kanto I	Tokyo	76	170
		Kanto II	Nagano	78	180
		Hokuriku	Kanazawa	77	180
		Kinki II	Nara	65	150
	2007	Kanto I	Yokohama	91	210
		Hokuriku	Niigata	94	210
		Shikoku	Kochi	79	180
	2008	Hokkaido	Sapporo	81	180
		Tohoku	Sendai	78	180
		Kinki I	Osaka	80	180
		Kita-Kyushu	Fukuoka	69	160
	2009	Kanto I	Tokyo	76	170
		Hokuriku	Kanazawa	72	160
		Chugoku	Hiroshima	85	190
		Shikoku	Kochi	79	180
	2010	Tokai	Nagoya	75	170
Mean ± SD *^2^				78.0 ± 7.3	178 ± 16
After the FDNPP *^1^ accident	2011	Tohoku	Sendai	88	200
		Tohoku	Fukushima	81	180
		Kanto I	Tokyo	78	180

*^1^ FDNPP: Fukushima Daiichi nuclear power plant. *^2^ Mean and standard deviation value before the Fukushima Daiichi nuclear power plant accident.

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
