# Peer review of "Total Diet Study to Assess Radioactive Cs and ^40^K Levels in the Japanese Population before and after the Fukushima Daiichi Nuclear Power Plant Accident"

_ijerph, 2020, doi:10.3390/ijerph17218131_

Round 1
Reviewer 1 Report
The manuscript " Total Diet Study to Assess Radioactive Cs ..." describes the results obtained by total diet studies performed in the years before the Fukushima Daiichi nuclear power plant (FDNPP) accident (2006-2010) and approximately six months after the accident. As the accident took place in 2011 I wonder why it took so long (9 years !!!) to make an efford to publish the data. In addition I wonder, why no data from stidies after 2011 have been included in the manuscript. Were no data obtained since then? I can not imagine. The manuscript would have profited considerably by a more recent study (let's say 2018-2019).
It has been certainly a lot of work to prepare and measure all these samples. However, the scientific output is very limited. The paper is more or less a mere descriptive attempt then a scientific paper, because some important information is missing:
1) no statistics (not even simple statistics) have been performed. Most tables show only the measurement range, not even means and standard deviations. That should be done.
2) The information on food supply ist mostly missing. For all the cities in Japan, it must be denied that food is produced in the cities! So the question is, where are the food items being produced. Will people from Tokyo and Kanazawa or Nagoya be supplied with food grown at the same agricultural area? More information is needed.
There are a few minor linguistic inaccuracies, which should be corrected.
1) The term survey is used frequently. In some cases, it would be better to be more accurate and use the term measurents or analysis.
2) Instead of the term "bounds" the word "limits" should be used.
However, there are also some phrases, which are not understandable or even misleading and wrong:
1) Line 50 "Initially, pesticides and their metabolites, including seven metals (Pb, Cd, total-Hg, total-As, Cu, Mn, and Zn) as well as total polychlorinated biphenyl (PCB) ..." . The word "including" refers to the preceding nouns. This declares the items listed in brackets as pesticies.
2) Line 103 "Activity concentrations of the three nuclides were decay-corrected to the end of the sampling period and expressed in Bq/kg of fresh weight." What is meant with ..to the end of the sampling period...? Not understandable what has been done.
3) Line 159 "When the activity concentrations were decay-corrected to the date of the FDNPP accident, the concentration ratios of ...". What do you want to say? People recieved at the day of the accident a certain amount of contamination? But... at that time food was not yet contaminated. In order to get contaminated food, the plants first have to enrich the nuclides and grow, they have to be harvested ... This takes time.
4) Line 241 "While the lower and upper bounds of radioactive Cs intake were significantly different in each MB before the FDNPP accident, they were almost the same after the accident because most of the TDS samples contained detectable levels of 137Cs and 134Cs." In no way understandable.
5) Line 298 " The minimum, maximum, and mean total CED values of 40K were 150 μSv, 210 μSv, and 180 μSv (SD = 16), respectively, which are in close agreement with a previous TDS [13]. The mean value obtained was approximately ten times higher than the maximum CED obtained for radioactive Cs in the present study, and was comparable to the worldwide average of 170 μSv [39]. " This suggest a realtionship between K and Cs, which is for sure not given.
Criticism of the tables
The tables should have the same layout.
Table 1) The table shows pooled data (n>1) for the different years. Accordingly, data should be given as means and SD and range (optional). What do you mean by "the percentage of detection"?
Table 2) The table does not have the same header. Add "Activity concentration of" above "137Cs (Bq/kg)....134Cs (Bq/kg)....37Cs + 134Cs (Bq/kg)
Table 3) The table shows pooled data (n>1) for the different years. Accordingly, data should be given as means and SD and range (optional) for each year. The column for 2011 is included in the column "Mean ±SD". I suggest to show "Mean ±SD (2006-2010" for the years before the accident on the left side from "2011" and add in the table legend (above the line showing the years) " before the ...accident" and "after ...the accident", just for clarity.
Table 4) The data for Cs and K should be presented in separate tables as there is no relationship between Cs and K. The data should be presented as means and SD and range (optional). Ther should be subheadings, such as "before the ...accident" and "after ...the accident", just for clarity.
Criticism of the figures
Figure 1) ok
Figure 2) what are the 3-4 symbols per year depicting? The different cities? I have the impression that the data should be visualized as histograms. Axis titles are to small. Use capital letters for first letter.
Figure 3) The histogram bars should be arranged based on the distance from the location of the accident. Distande should be added to the X-axis (together with city name). Food items should be sorted according to their appearance in the tables (rice on top and drinking water down) for bars and legend.
Figure 4) see figure 3
Author Response
Dear reviewer,
Thank you for taking the time to review our work.
We trust that all of your comments have been addressed accordingly in a revised manuscript.
Please see the attachment.
Thank you so much for your effort.
Best regards,
Hiroshi Terada
Department of Environmental Health, National Institute of Public Health

Reviewer 2 Report
Please find my comments and suggestions in attached file.

Author Response

(The authors gave the same response as above.)

Round 2
Reviewer 1 Report
The resubmitted manuscript shows considerable improvements. I am fine with it